# Maternal Diet High in Linoleic Acid Alters Offspring Lipids and Hepatic Regulators of Lipid Metabolism in an Adolescent Rat Model

**DOI:** 10.3390/ijms25021129

**Published:** 2024-01-17

**Authors:** Nirajan Shrestha, Simone L. Sleep, Olivia J. Holland, Josif Vidimce, Andrew C. Bulmer, James S. M. Cuffe, Anthony V. Perkins, Andrew J. McAinch, Deanne H. Hryciw

**Affiliations:** 1School of Pharmacy and Medical Science, Griffith University, Southport, QLD 4222, Australia; nirajan.shrestha@griffithuni.edu.au (N.S.); simone.sleep@griffithuni.edu.au (S.L.S.); o.holland@griffith.edu.au (O.J.H.); josif.vidimce@griffithuni.edu.au (J.V.); a.bulmer@griffith.edu.au (A.C.B.); a.perkins@griffith.edu.au (A.V.P.); 2Women’s, Newborn and Childrens Services, Gold Coast Health, Southport, QLD 4222, Australia; 3School of Biomedical Sciences, The University of Queensland, St Lucia, QLD 4072, Australia; j.cuffe1@uq.edu.au; 4School of Health, University of Sunshine Coast, Sunshine Coast, Sippy Downs, QLD 4556, Australia; 5Institute for Health and Sport, Victoria University, Melbourne, VIC 3001, Australia; andrew.mcainch@vu.edu.au; 6Australian Institute for Musculoskeletal Science (AIMSS), Victoria University, St. Albans, VIC 3021, Australia; 7School of Environment and Science, Griffith University, Nathan, QLD 4111, Australia; 8Griffith Institute of Drug Discovery, Griffith University, Nathan, QLD 4111, Australia

**Keywords:** linoleic acid, maternal, offspring, sex-specific liver

## Abstract

Linoleic acid (LA), an n-6 polyunsaturated fatty acid (PUFA), is essential for fetal growth and development. A maternal high LA (HLA) diet alters cardiovascular development in adolescent rats and hepatic function in adult rats in a sex-specific manner. We investigated the effects of an HLA diet on adolescent offspring hepatic lipids and hepatic lipid metabolism gene expression, and the ability of the postnatal diet to alter these effects. Female Wistar Kyoto rats were fed low LA (LLA; 1.44% energy from LA) or high LA (HLA; 6.21% energy from LA) diets during pregnancy and gestation/lactation. Offspring, weaned at postnatal day (PN) 25, were fed LLA or HLA and euthanised at PN40 (*n* = 6–8). Maternal HLA increased circulating uric acid, decreased hepatic cholesterol and increased hepatic *Pparg* in males, whereas only hepatic *Srebf1* and *Hmgcr* increased in females. Postnatal (post-weaning) HLA decreased liver weight (% body weight) and increased hepatic *Hmgcr* in males, and decreased hepatic triglycerides in females. Maternal and postnatal HLA had an interaction effect on *Lpl*, *Cpt1a* and *Pparg* in females. These findings suggest that an HLA diet both during and after pregnancy should be avoided to improve offspring disease risk.

## 1. Introduction

Fatty acids (FAs) represent an important source of energy in periods of catabolic stress [1] their oxidation produces acetyl-CoA, which supplies energy to other tissues when glycogen stores are depleted. However, dysregulation of mitochondrial oxidation of FAs can contribute to the development of lipid dysfunction in liver. The accumulation of lipids in the liver is deleterious to health, and is thought to be due to an imbalance between lipid availability in the diet and the capacity of the liver to dispose of excessive lipids caused by an increased intake in the diet [2]. Research has demonstrated that consumption of a high fat maternal diet, both before and during pregnancy, plays a significant role in the development of fatty liver and associated diseases in offspring [3]. What currently remains unknown are the effects of specific types of fat and fatty acids on offspring disease. Consumption of the polyunsaturated fatty acid, linoleic acid (LA, an essential n-6 fatty acid), has increased in recent years [4]. LA, an essential FA, which can only be obtained in the diet [4], can be metabolised to form inflammatory mediators [5].

As women of childbearing age are consuming elevated concentrations of LA [4], research is warranted to investigate if during critical periods of development in utero, exposure to elevated maternal LA may be contributing to the increase in diseases, including liver disease. Previous research has demonstrated that adult offspring (postnatal (PN) 180) from mothers consuming an LA rich diet (12.3% LA) have increased triglyceride accumulation in hepatocytes and altered hepatic lipid metabolism that may predispose the offspring to the development of liver disease later in life [6]. In Western cultures [4], these concentrations are higher than what is typically consumed. Recently, we reported an increase in maternal liver weight and changes in inflammatory cytokines in the maternal liver of rats fed with elevated LA [7] at the same LA concentration that Australians are consuming (~6.21%) [4]. Furthermore, elevated maternal LA results in elevated circulating LA in the fetus [7]. However, the effects of an elevated maternal LA diet on offspring health have had limited investigations.

Developmental programming of disease risk due to an adverse maternal environment during pregnancy has led to the identification of sex-specific effects on offspring outcomes [8,9,10,11,12,13]. To add to this field of study, we have investigated the effects of a maternal HLA diet on rodent offspring lipid metabolism, and circulating lipids and fatty acids in adult offspring [12]. We have shown that a maternal HLA diet decreased total cholesterol, HDL-cholesterol and triglycerides in the plasma of rodent male offspring [12]. Further, the maternal HLA diet was associated with the downregulation in hepatic *Hmgcr* in both male and female rodent offspring, and the downregulation in hepatic *Cpt1a* and *Acox1* in rodent females. These targets are critical to FA oxidation in the liver and may contribute to changes in circulating FA concentrations.

There are windows of development, with both the maternal and postnatal (post-weaning) metabolic trajectories having different impacts on metabolism [12]. Adolescence is a critical period of development. We have previously investigated the effect of a maternal high linoleic acid (HLA) diet on circulating lipids, fatty acids, cardiovascular function in adolescent (PN40) offspring [14]. In that study, we demonstrated circulating total cholesterol and HDL-cholesterol in females and decreased total plasma n-3 FA in males, while maternal and postnatal HLA diets decreased total plasma n-3 FA in females. The n-3 α-linolenic acid (ALA) and eicosapentaenoic acid (EPA) were decreased by postnatal but not maternal HLA diets in both sexes, and maternal and postnatal HLA diets increased total plasma n-6 and LA, and a maternal HLA diet led to increased circulating leptin, in both male and female offspring [14]. This study also demonstrated that there were sex-specific changes in cardiovascular function in adolescent offspring. Notably, the restoration of LA concentrations with the LLA diet in the postnatal period reduced the ratio of LA:ALA but did not improve cardiac outcomes [14]. At this time the hepatic lipid metabolism in adolescent offspring has not been investigated.

The postnatal environment is a potential window to modify disease phenotypes caused by an adverse maternal environment [14]. Research in this field has mainly focused on outcomes associated with a maternal obesogenic diet. For example, adverse metabolic outcomes associated with a maternal obesogenic diet could not be restored with a control postnatal diet; however, a postnatal obesogenic diet further exacerbated these adverse effects of maternal obesity [15]. Given that fetal programming describes how the fetus adapts to the dietary factors it has been exposed to, changing the postnatal environment to one that is different from that experience in utero may contribute to adult-onset of disease. At this time, we do not know if the impact of an elevated maternal LA diet on offspring health could be reversed by a postnatal diet with recommended concentrations of LA or if this may instead exacerbate negative outcomes.

In this study, we aimed to investigate the effects of a maternal HLA diet on lipid metabolism in the liver of adolescent (pre-puberty) rat offspring. Furthermore, we investigated whether the postnatal diet could reverse or exacerbate any adverse effects of the maternal diet high in LA. As developmental programming is often sex-specific [8,9,10,11,12,13], we analysed both male and female adolescent offspring independently.

## 2. Results

### 2.1. Effect of Maternal and Postnatal HLA Diet on Liver Weight

Organ and body weight from the PN40 cohort has been previously published [14]. To extend this study, we investigated liver weight in this cohort. A postnatal HLA diet decreased liver weight (% body weight) in males (*n* = 6–8, *p* = 0.0148) (Table 1). There was no change in liver weight due to maternal or postnatal HLA in females.

### 2.2. Effect of Maternal and Postnatal HLA Diet on Circulating Liver Enzymes

There were no changes to circulating liver enzymes alanine amino-transferase (ALT), serum aspartate (AST), alkaline phosphatase (ALP), total bilirubin and urea in response to a maternal or postnatal HLA. Maternal HLA diet increased uric acid in males (*n* = 6–8, *p* = 0.022) and there was an interaction effect for uric acid in females (*n* = 6–8, *p* = 0.025) (Table 2).

### 2.3. Effect of Maternal and Postnatal HLA Diet on Hepatic Lipids in Adolescent Offspring

A maternal HLA diet decreased hepatic cholesterol in males (*n* = 6–8, *p* = 0.019) but did not alter hepatic cholesterol in females. A postnatal HLA diet did not alter hepatic triglycerides in females but decreased hepatic triglycerides in females (*n* = 6–8, *p* = 0.047) (Figure 1).

### 2.4. Effect of Maternal and Postnatal HLA Diet on Hepatic Lipids in Adolescent Offspring

*Ldlr* was unaltered in male and female offspring (*n* = 6–8, Figure 2). A maternal HLA diet increased *Srebf1* (*p* = 0.0128) and *Hmgcr* (*n* = 6–8, *p* = 0.002) in female offspring (Figure 2). Postnatal HLA increased *Hmgcr* in males (*n* = 6–8, *p* = 0.029) (Figure 2). There was an interaction effect for *Lpl* in females (*n* = 6–8, *p* = 0.0171) (Figure 2).

### 2.5. Effect of Maternal and Postnatal HLA Diet on Genes Responsible for Fatty Acid Oxidation

*Cyp27a1* and *Acox1* were unaltered in male and female offspring. A maternal HLA diet increased *Pparg* (*n* = 6–8, *p* = 0.01) in male offspring (Figure 3). There was an interaction effect for *Pparg* in females (*n* = 6–8, *p* = 0.019) (Figure 3).

## 3. Discussion

Chronic liver disease is increasing in incidence due to its association with metabolic syndrome and its associated pathologies [16]. In the present study, we investigated the effect of a maternal diet high in LA on hepatic lipid metabolism and hepatic expression of genes responsible for lipid metabolism and fatty acid oxidation in adolescent (PN40) rat offspring. Further, we evaluated if modification of the postnatal diet to a control LLA could reverse any adverse phenotypes that were associated with the HLA maternal diet. In this study, we showed sex-specific effects of a maternal HLA in offspring hepatic function. Specifically, this study showed that the maternal HLA diet altered hepatic cholesterol, hepatic lipid metabolism gene expression (*Srebf1*, *Hmgcr*) and a hepatic fatty acid oxidation gene expression (*Pparg*) in a sex-specific manner. A postnatal HLA diet altered hepatic triglycerides and the expression of a hepatic lipid metabolism gene (*Hmgcr*) in a sex-specific manner. There was also an interaction affect for hepatic lipid metabolism gene expression (*Lpl*) and a hepatic fatty acid oxidation gene (*Pparg*) in a sex-specific manner. Our previous study demonstrated that a maternal HLA diet significantly decreased both total cholesterol and HDL cholesterol in the plasma of female but not male offspring at PN40 [14]. Collectively, these data suggest that in male offspring, a maternal HLA diet alters cholesterol storage via *PParg* and may alter FA storage in the liver. The postnatal diet alters cholesterol and triglyceride handling in the liver in a sex-specific manner. Our previous research on these cohorts demonstrated that the maternal and postnatal high LA diet did not change total plasma saturated FA, but maternal HLA diet did increase trans-FA in female offspring [14]. In both sexes, both maternal and postnatal HLA diet significantly altered total plasma monounsaturated FA [14]. These changes may be mediated by the changes identified in hepatic targets associated with fatty acid oxidation and metabolism.

Despite the exposure to a lower LA diet only lasting ~15 days in the current study, postnatal effects and interactive effects of HLA were observed. We have previously demonstrated that the change in LA in the diet for 15 days postweaning alters circulating fatty acids in both sexes and cardiac function in female offspring [14]. Further, in rodents, liver mass increases significantly in the first postnatal weeks until hepatocyte proliferation stops at around ~PN30 [17]. The adult hepatocyte is a quiescent organ, with a very long half-life (~180 days in the mouse) and minimal replicative activity so that most cells are not mitotically active [17]. Thus the lack of change in liver weight at PN180 [12] does not imply that the offspring liver is not affected by the maternal diet. Changes in cholesterol, triglyceride or fatty acid handling are likely to be further exacerbated when additional nutritional stressors, such as a high fat diet, may contribute to a decline in hepatic function.

Fatty acids represent an important source of energy in periods of catabolic stress [1] their oxidation produces acetyl-CoA, which supplies energy to other tissues when glycogen stores are depleted. However, dysregulation of mitochondrial oxidation of FA plays a vital role in development of lipid dysfunction in liver. The liver is the most active organ for lipid metabolism in the body. Genes involved in liver lipid metabolism, such as *Lpl*, *Srebp-1*, *Pparg*, are extensively characterised cytoplasmic mediators in hepatic cells [18]. These targets were all altered in response to a high LA diet, in a sex-specific, and window-specific (pregnancy vs. postnatal) period in adolescent rats, which differs from what we have previously demonstrated in adult offspring [12]. *Lpl* critically modulates fatty acid metabolism by providing fatty acids and monoacylglycerol through catabolism of triacylglycerol within chylomicrons and very low density lipoproteins [18]. *Srebp-1*, which is a transcriptional regulator that regulates lipid synthesis, translocates to the nucleus following activation [19]. *Pparg*, a nuclear receptor, regulates lipid homeostasis [20]. Our results suggested that there are sex-specific differences in the modulation of *Lpl* and *Srebp1*. Studies have shown that long-term intake of a diet high in fat increases the expression levels of genes related to fat synthesis in the liver, such as *Srebp1*, and decreases the expression levels of genes related to fat decomposition, such as *Pparg*. This significantly increases hepatic total fat, triglycerides and free fatty acids content [21].

Recent literature suggests that the dietary imbalance of high n-6 and low n-3 PUFA intake, characteristic of the Western diet, is closely associated with the development of non-alcoholic fatty acid liver disease (NAFLD), especially in adolescents [22]. Research has demonstrated beneficial outcomes associated with a reduction in the n-6/n-3 ratio for the risk of NAFLD [23]. In addition to diet, research has suggested that genetic polymorphisms in important genes responsible for lipid homeostasis could contribute to disease risk. For example, a polymorphic variant in the patatin-like containing domain phospholipase 3 (*PNPLA3*) I148M variant is the major genetic risk factor for all stages of fatty liver disease via hepatic mitochondrial dysfunction [24]. Further, the *PNPLA3* rs738409 gene polymorphism, which alters adipocyte size [25], enhances the benefits of a reduced n-6/n-3 diet in obese adolescents [23]. Thus, improvements in hepatic function due to beneficial changes in PUFA may be augmented by the genetic background of the individual.

This study also showed increased serum uric acid in males in response to maternal HLA, and an interaction effect for females. Hyperuricemia has long been associated with high-fat diets [26], and can contribute to dyslipidemia and metabolic syndrome [27]. Interestingly, hyperuricemia is associated with chronic kidney disease [28]. Further, uric acid induces immune system activation and alters the characteristics of kidney cells so that they are proinflammatory and profibrotic [28]. It is well known that consuming a high-fat diet results in fat accumulation, increased inflammatory cytokines, induction of glomerular retraction and renal dysfunction [29]. The concentrations of uric acid in these rodents are within the normal physiological range [30]. However, as uric acid is handled by the kidney, what remains unknown are the direct effects of a maternal HLA diet on renal function in offspring, which requires further investigation.

## 4. Materials and Methods

### 4.1. Experimental Animal Model and Diet

Ethical approval was granted by the Griffith University Animal Ethics Committee (NSC/01/17/AEC: 26 April 2017). Wistar Kyoto rats (8 weeks of age; *n* = 8 for diet with low linoleic acid (LLA) and *n* = 10 for HLA diet) were purchased from the Australian Resource Centre (ARC, Kensington, Australia) and housed in accordance with the Australian Code of Practice for Care and Use of Animals for Scientific Purpose, following the ARRIVE Guidelines for Reporting Animal Research [31].

Eight-week-old female Wistar Kyoto (WKY) rats were housed in individually ventilated cages under 12 h light-dark cycles at a temperature of 20–22 °C, and were provided with standard food pellets during acclimatisation and tap water ad libitum throughout the study. After a week for acclimatization, female rats were randomised to consume either a control low LA (LLA: 1.44% of energy from LA, *n* = 8) or high LA (HLA: 6.21% of energy from LA, *n* = 8) diet for 10 weeks, then mated with a male rat consuming a chow diet. The minimum requirement for LA in the rodent diet is between 1–1.5% [32]. The composition of the custom diet has been previously reported [7]. These diets were isocaloric and matched for n-3 PUFA and total fat content. Mothers consumed the same diet during pregnancy and lactation. Offspring from mothers were weaned at PN25 and then randomly allocated to a group to be fed with either a LLA or HLA diet. This created the following groups: LLA-LLA, LLA-HLA, HLA-LLA, HLA-HLA (where the first diet is maternal and the second is postnatal (post-weaning)). At sacrifice, blood samples were collected by cardiac puncture, centrifuged at 5000× *g* for 10 min to separate plasma, and stored at −80 °C for analysis. Offspring were terminally anesthetised with an intraperitoneal injection of sodium pentobarbital (60 mg/kg) at PN40 prior to the onset of puberty. Organs were weighed and snap-frozen in liquid nitrogen, then stored at −80 °C until downstream analysis. Organ weights from this study have previously been published [14].

### 4.2. Cholesterol and Triglyceride Quantification in the Liver of Offspring and Plasma Biochemical Parameters

Lipids from the liver tissue were extracted following previously published methods [33]. Liver tissue was homogenised while frozen using a mortar and pestle. Lipids from the homogenized sample were extracted by two sequential solvent extractions with isopropanol. During each extraction, homogenates were vortexed, sonicated for 10 min and then centrifuged at 43,000× *g* for 10 min. The two supernatant fractions were collected in new 15 mL falcon tubes and evaporated using the Roto—evaporator (Maxivac, Labogene, Bjarkesvej, Lillerød, Denmark) at 35 °C (500× *g*, 1 kPa pressure). The remaining dry pellet was reconstituted in 250 µL of isopropanol and loaded into an automated biochemistry analyser (COBAS Integra 400+, Roche Diagnostics, North Ryde, Australia) for the quantification of total cholesterol and triglyceride. Total cholesterol and triglyceride kits were purchased from Roche Diagnostics. These kits were verified with their appropriate calibrators and quality control (QC) prior to sample analysis. Isopropanol did not interfere with cholesterol or triglyceride contents.

Plasma biochemical parameters were assessed using an automated chemistry analyser using the manufacturers protocol (Integra 400 plus, Roche Diagnostics). All biochemistry assays were performed using Roche certified assay kits, which were calibrated using Calibrator for Automated Systems reagent. Quality control standards (PreciControl ClinChem Multi 1 and 2; Roche Diagnostics) were run prior to sample analysis to ensure accuracy of results. All analyses were performed in duplicate.

### 4.3. Quantitative Real Time Polymerase Chain Reaction (qPCR)

Total RNA was extracted from liver tissue using an RNeasy Mini kit (Qiagen, Chadstone, Australia) following the manufacturer’s guidelines. The quantification and evaluation of purity of RNA samples were assessed using the NanoDrop 1000 spectrophotometer (Thermo Fisher Scientific, Waltham, MA, USA). Reverse transcription of RNA to synthesize complementary DNA was performed using the iScript gDNA clear cDNA synthesis kit (BioRad, Hercules, CA, USA) following manufacturer’s guidelines. Quantitative PCR was performed using QuantiNova SYBR^®^ green master mix (Qiagen) following the manufacturer’s guidelines, in line with the Minimum Information for Publication of Quantitative Real-Time PCR Experiments (MIQE) guidelines [34]. All the primers used for this study were KiCqStart^TM^ predesigned primers from Sigma-Aldrich (St. Louis, MO, USA). PCR initial heat activation was run for 2 min at 95 °C, then qPCR reactions were run for 40 cycles of 95 °C for 5 s (denaturation) and 60 °C for 10 s (combined annealing/extension) using StepOne^TM^ real-time PCR systems (Applied Biosystems, Waltham, MA, USA). Samples were run in duplicate, and the average of the two was used in analysis. Gene expression was quantified using the 2^−ΔΔ*C*q^ method normalised to the geometric mean of β-actin and β-2 microglobulin as reference genes. These reference genes were stable across the treatment groups.

### 4.4. Statistical Analysis

All data were analysed using GraphPad Prism 8.3.1. One male and one female offspring from each litter were analysed. *n* values represent individual offspring from separate litters. Data were analysed separately for males and females, with each sex analysed by two-way ANOVA with maternal and postnatal diets as the factors. Specific comparisons were made using Tukey post hoc test. Data are presented as mean ± standard error of the mean (SEM). *p*-values < 0.05 were considered evidence of significant differences.

## 5. Conclusions

In conclusion, we have demonstrated that a maternal and postnatal (adolescent) HLA diet alters lipid accumulation and hepatic expression of genes involved in hepatic handling of lipid storage, fatty acid storage and lipogenesis. These findings suggest that maternal or postnatal HLA diets are independent factors for offspring hepatic function, with this study demonstrating changes in key genes responsible for lipid metabolism and fatty acid oxidation occurring in adolescence. As sex-specific differences were observed in the current study, future research should investigate both sexes for the effect on hepatic health. Further studies should be conducted to identify the exact mechanism of altered lipid metabolism in the offspring liver due to maternal HLA diet.

## Figures and Tables

**Figure 1 ijms-25-01129-f001:**
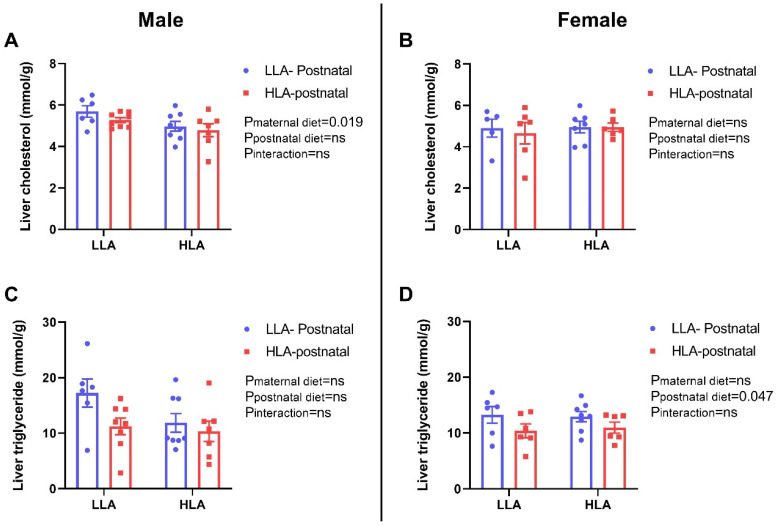
Effect of maternal or postnatal diet high in linoleic acid on cholesterol and triglyceride contents in the liver of adolescent offspring. (**A**,**B**) liver cholesterol (**C**,**D**) liver triglycerides, (**A,C**) male and (**B,D**) female. Data are presented as mean ± standard error of the mean (SEM). Two-way ANOVA was performed for statistical analysis with maternal diet and postnatal diet as two factors. *n* = 6–8. LLA: low linoleic acid; HLA: high linoleic acid. ns: non-significant.

**Figure 2 ijms-25-01129-f002:**
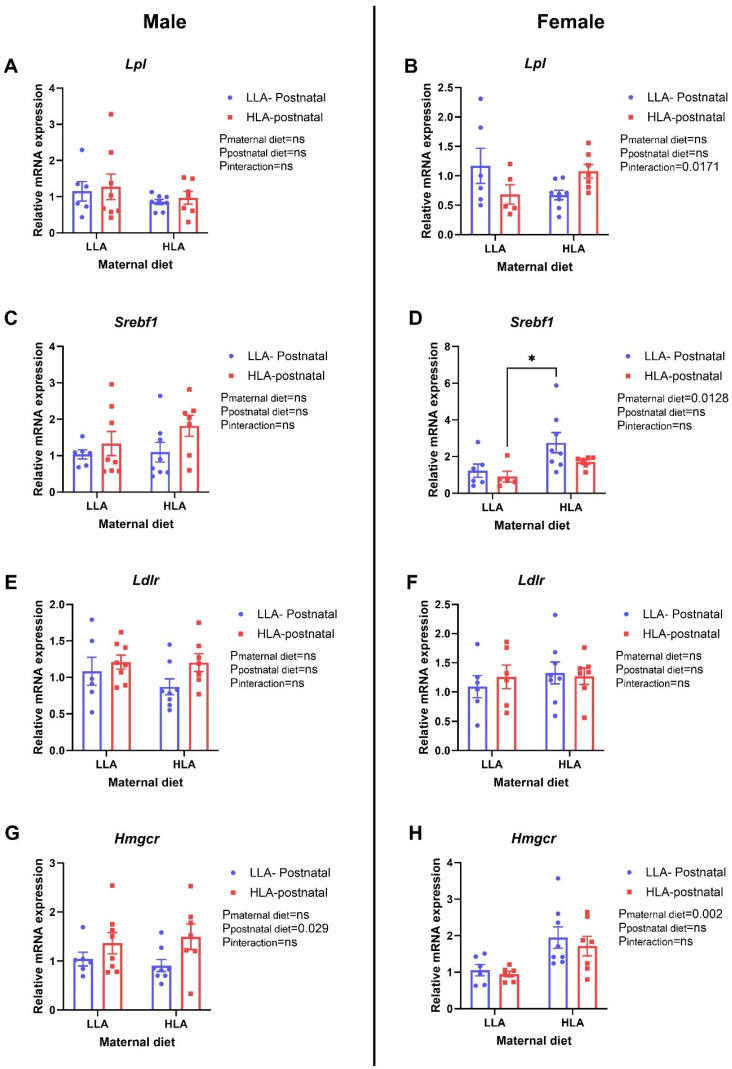
Effect of maternal or postnatal diet high in linoleic acid on expression of genes related to lipid metabolism. (**A**,**B**) *Lpl*, (**C**,**D**) *Srebf1*, (**E**,**F**) *Ldlr*, (**G**,**H**) *Hmgcr*. (**A**,**C**,**E**,**G**) are for males and (**B**,**D**,**F**,**H**) are for females. Data are presented as mean ± standard error of the mean (SEM). Two-way ANOVA was performed for statistical analysis with maternal diet and postnatal diet as two factors. *n* = 6–8. LLA: low linoleic acid; HLA: high linoleic acid. Where post hoc analysis identified a difference, differences across the groups are denoted by an asterix (* *p* < 0.05). ns: non-significant.

**Figure 3 ijms-25-01129-f003:**
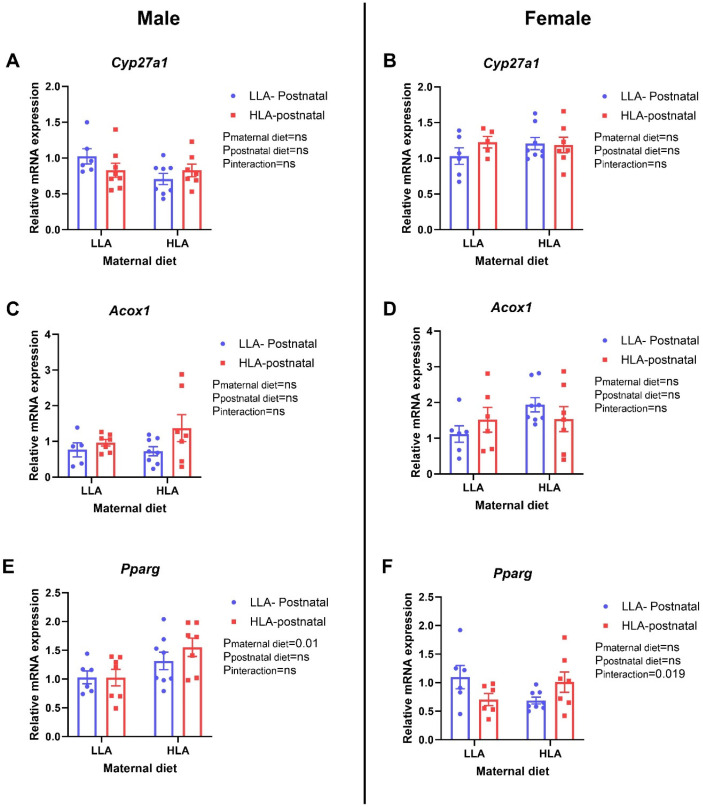
Effect of maternal or postnatal diet high in linoleic acid on expression of genes related to fatty acid oxidation. (**A**,**B**) *Cyp27a1*, (**C**,**D**) *Acox1*, (**E**,**F**) *Pparg*. (**A**,**C**,**E**) are for males and (**B**,**D**,**F**) are for females. Data are presented as mean ± standard error of the mean (SEM). Two-way ANOVA was performed for statistical analysis with maternal diet and postnatal diet as two factors. *n* = 6–8. LLA: low linoleic acid; HLA: high linoleic acid. ns: non-significant.

**Table 1 ijms-25-01129-t001:** Effect of maternal or postnatal diet high in linoleic acid on offspring (PN40) liver weight.

	LLA Maternal Diet		HLA Maternal Diet		*p*-Values		
Parameters	LLA PN Diet	HLA PN Diet	LLA PN Diet	HLA PN Diet	Maternal Diet	Postnatal Diet	Interaction
Male							
% Liver	4.56 ± 0.16	4.26 ± 0.09	4.40 ± 0.06	4.21 ± 0.04	ns	0.0148	ns
Female							
% Liver	4.65 ± 0.15	4.51 ± 0.12	4.63 ± 0.08	4.44 ± 0.1	ns	ns	ns

LLA = low linoleic acid, HLA = high linoleic acid, PN = postnatal (post-weaning), ns = not significant.

**Table 2 ijms-25-01129-t002:** Liver enzymes, urea and uric acid levels in the plasma of adolescent offspring.

	LLA Maternal Diet		HLA Maternal Diet		*p*-Values		
Parameters	LLA PN Diet	HLA PN Diet	LLA PN Diet	HLA PN Diet	Maternal Diet	PN Diet	Interaction
Male							
ALT (U/L)	50.9 ± 4.53	42.2 ± 3.01	45.7 ± 3.61	47.8 ± 4.27	ns	ns	ns
AST (U/L)	153.3 ± 13.8	140.1 ± 14.7	131.6 ± 7.9	150.1 ± 22.1	ns	ns	ns
ALP (U/L)	208.8 ± 35.9	227.8 ± 44.2	217.3 ± 33.8	218.4 ± 53.7	ns	ns	ns
Total bilirubin (μM/L)	1.16 ± 0.08	1.01 ± 0.11	1.08 ± 0.10	1.20 ± 0.21	ns	ns	ns
Urea (mM/L)	5.05 ± 0.40	4.96 ± 0.33	5.08 ± 0.41	4.70 ± 0.36	ns	ns	ns
Uric acid (μM/L)	53.5 ± 1.99 ^a^	53.5 ± 4.97 ^a,b^	72.7 ± 6.01 ^b^	62.5 ± 7.30 ^a,b^	0.022	ns	ns
Female							
ALT (U/L)	41.8 ± 3.92	38.9 ± 3.58	37.5 ± 1.04	43.0 ± 3.54	ns	ns	ns
AST (U/L)	132.0 ± 10.9	124.5 ± 13.5	139.9 ± 17.2	136.8 ± 10.7	ns	ns	ns
ALP (U/L)	157.3 ± 23.5	217.3 ± 38.8	203.8 ± 33.3	184.3 ± 17.0	ns	ns	ns
Total bilirubin (μM/L)	0.79 ± 0.09	0.88 ± 0.19	0.88 ± 0.11	0.76 ± 0.09	ns	ns	ns
Urea (mM/L)	4.88 ± 0.58	5.99 ± 0.10	5.26 ± 0.29	5.47 ± 0.24	ns	ns	ns
Uric acid (μM/L)	55.5 ± 5.93 ^a,b^	40.2 ± 3.88 ^a^	52.8 ± 2.71 ^a,b^	65.9 ± 9.29 ^b^	ns	ns	0.025

Alanine amino-transferase = ALT, serum aspartate = AST, alkaline phosphatase = ALP, LLA = low linoleic acid, HLA = high linoleic acid, PN = postnatal (post-weaning), ns = not significant. Where post hoc analysis identified a difference, differences across the groups are denoted by a different letter.

## Data Availability

Data are contained within the article.

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
