# Peer review of "Maternal Diet High in Linoleic Acid Alters Offspring Lipids and Hepatic Regulators of Lipid Metabolism in an Adolescent Rat Model"

_ijms, 2024, doi:10.3390/ijms25021129_

Round 1

Reviewer 1 Report

Comments and Suggestions for Authors

Some points should be addressed before the manuscript can be considered for publication.

The main concerns are as follow:

1.     Please provide a comprehensive experimental scheme, detailing how animals transitioned between maternal and postnatal diets. It is crucial to specify what is meant by 'postnatal' in this context – does it refer to the period following parturition or after weaning? If it refers to the period post-parturition, clarify how mothers with their pups were transitioned from one dietary treatment to another. Was this transition randomized? Detailed information in this regard will enhance the understanding of the experimental design and the contextual relevance of the results.

2.     For statistically significant interactions, it is important to present the results of the Tukey test for all dietary treatments. This approach is necessary to clearly demonstrate which groups differ from each other. For instance, in Table 2, results for uric acid in females should include comparisons among all groups, not just arbitrarily selected ones. This requirement extends to other cases such as Lpl for females (Figure 2), Cpt1a for both males and females, and Pparg for females (Figure 3).

3.     It's essential to confirm whether the data were checked for compliance with ANOVA assumptions, particularly concerning the homogeneity of variances. The datapoints presented in the figures suggest that for some cases, especially in relation to the assumption of homogeneity of variances, there might have been violations.

Minor comments:

L47: "As women of childbearing age are consuming elevated concentrations of LA” –Consider providing supporting data or references to substantiate this claim for added clarity and credibility.

L54: Clarify how the typical consumption patterns in Western cultures relate directly to the study's objectives.

L55: The transition to discussing the effects of elevated maternal LA is somewhat abrupt.

L51: "For clarity, explicitly state that 'PN180' refers to the age of rat offspring in rat-based study.

L64: Define the HLA abbreviation at its first mention to ensure clarity for the readers.

L73: Again, specify that this was a rat study.

L100: Please specify, that experiment ended and rats were killed before they reached sexual maturity (6-8th week of life).

L133 and elsewhere in the manuscript: verify that gene names are italicized

L223: The discussion on increased serum uric acid and its implications is intriguing. However, it would be beneficial to clarify if these results are still within the physiological reference ranges for adolescent rats. This information is crucial for interpreting the significance and potential health impacts of these findings.

L247: Was pregnancy included in this 10-week-long period, or were dams matched after this period and dietary treatments continued?

L249: Some basic information about the effect of maternal diets on pregnancy outcomes (litter size, animal weight, male-to-female ratio, etc.) is required.

L252: Provide a detailed list of the organs sampled (just liver?) and include information about the blood collection process.

L269: The description of plasma biochemical parameter assessment is thorough.

L276: Include primer sequences and rationale for the selection of specific genes for analysis.

L289: Specify the number of technical replicates used in qPCR to assess the robustness of the data.

Author Response

We appreciate the comments from the reviewer and believe the revision is improved because of their suggestions. 

Reviewer 2 Report

Comments and Suggestions for Authors

The effects of maternal milk rich in linoleic acid are studied in this article in rats, comparing different groups with different treaments. Many genes involved in liver metabolism, as well as levels of different lipids have been measured for each group.

As I minor comment I thhink that all Tables need a legend with all the abbreviations used.

Overall, I consider that this article has few experiments and few data, althought the Introduction is well-explained and the Discussion includes proper citations. 

I would appreciate more mechanistic insights into the role of each gene expressed and each molecule present in the liver, to find a pathway linking the effect of oleic acid with a cellullar of molecular mechanism.

Therefore, it think that is does not meeting the scope and the scientific standards of the journal.

Author Response

(The authors gave the same response as above.)

Round 2

Reviewer 2 Report

Comments and Suggestions for Authors

Thank you for addressing the issues that I highlighted.